# Evaluating biological plausibility of learning algorithms the lazy way

**Owen Marschall**
Center for Neural Science
New York University
New York, NY 10003
oem214@nyu.edu

**Kyunghyun Cho**
New York University
Facebook AI Research
CIFAR Azrieli Global Scholar
kyunghyun.cho@nyu.edu

**Cristina Savin**
Center for Neural Science
Center for Data Science
New York University
csavin@nyu.edu

## Abstract

To which extent can successful machine learning inform our understanding of biological learning? One popular avenue of inquiry in recent years has been to directly map such algorithms into a realistic circuit implementation. Here we focus on learning in recurrent networks and investigate a range of learning algorithms. Our approach decomposes them into their computational building blocks and discusses their abstract potential as biological operations. This alternative strategy provides a "lazy" but principled way of evaluating ML ideas in terms of their biological plausibility.

The phenomenal success of neurally inspired machine learning (ML) algorithms has captured the imagination of many neuroscientists hoping to understand the principles of learning in the brain. This led to renewed efforts to find biologically realistic approximations to the backpropagation algorithm, the key to efficient credit assignment in feedforward networks [1, 2, 3, 4, 5]. Relatively little work covers the—arguably more realistic—scenario of learning in recurrent networks and *temporal* credit assignment [6]. The canonical machine learning solution, **backpropagation through time** (BPTT, [7]), poses more severe challenges when attempting to map it to biology. In particular, BPTT is *temporally nonlocal*; that is, each weight update depends on network activity across multiple time points. This problem could be addressed by using **online** alternatives to BPTT such as Real-Time Recurrent Learning (RTRL, [8]). Unfortunately, RTRL has an $\mathcal{O}(n^3)$ memory requirement (for $n$ neurons) and it is hard to imagine any biological structure for storing this many real-valued variables. The cubic complexity is a problem not only for biology, but also for machine learning and has precluded the use of RTRL in applications. Nonetheless, the practical need for efficient online learning has revived the RTRL idea, leading to several new algorithms [9, 10, 11, 12, 13], which all reduce memory complexity to $\mathcal{O}(n^2)$, but differ in the nature of their approximations [14]. Our goal here is to investigate to what extent these ideas could be implemented in a biological circuit.

One *could* take each algorithm individually and try to model in detail a biophysical implementation, à la [1, 2, 3, 4, 5]. However, it's unlikely that any single ML solution maps one-to-one onto neural circuitry. Instead, a more useful exercise would be to identify core computational building blocks that are strictly necessary for solving temporal credit assignment, which are more likely to have a direct biological analogue. To this end, we put forward a principled framework for evaluating biological plausibility in terms of the mathematical operations required–hence our "lazy" analysis. We examine several online algorithms within this framework, identifying potential issues common across algorithms, for example the need to physically represent the Jacobian of the network dynamics. We propose some novel solutions to this and other issues and in the process articulate biological mechanisms that could facilitate these solutions. Finally, we empirically validate that these biologically realistic approximations still solve temporal credit assignment, in two simple synthetic tasks.

**Plausibility criteria for recurrent learning.** Consider a recurrent network of $n$ units, with voltages $\mathbf{v}^{(t)} = \mathbf{W}\hat{\mathbf{r}}^{(t-1)}$, where $\hat{r}^{(t)}$ is the concatenation of recurrent and external inputs, with an additional constant input for the bias term, $\hat{\mathbf{r}}^{(t-1)} = [\mathbf{r}^{(t-1)}; \mathbf{x}^{(t)}; 1] \in \mathbb{R}^m$ ($m = n + n_{\text{in}} + 1$) and trainable weights organized as $\mathbf{W} = [\mathbf{W}^{\text{rec}}, \mathbf{W}^{\text{in}}, \mathbf{b}^{\text{rec}}] \in \mathbb{R}^{n \times m}$. For a closer match to neural circuits, the firing rates update in *continuous* time, via $\mathbf{r}^{(t)} = (1-\alpha)\mathbf{r}^{(t-1)} + \alpha\phi(\mathbf{v}^{(t)})$, using a point-wise neural activation function $\phi : \mathbb{R}^n \to \mathbb{R}^n$ (e.g. $\tanh$) and the network's inverse time constant $\alpha \in (0, 1]$. The network output $\mathbf{y}^{(t)} = \text{softmax}(\mathbf{W}^{\text{out}}\mathbf{r}^{(t)} + \mathbf{b}^{\text{out}}) \in \mathbb{R}^{n_{\text{out}}}$ is computed by output weights/bias $\mathbf{W}^{\text{out}} \in \mathbb{R}^{n_{\text{out}} \times n}$, $\mathbf{b}^{\text{out}} \in \mathbb{R}^{n_{\text{out}}}$ and compared with the training label $\mathbf{y}^{*(t)}$ to produce an instantaneous loss $L^{(t)}$.

BPTT and RTRL each provide a method for calculating the gradient of each instantaneous loss $\partial L^{(t)}/\partial W_{ij}$, to be used for gradient descent. BPTT unrolls the network over time and performs backpropagation as if on a feedforward network:

$$\frac{\partial L^{(t)}}{\partial W_{ij}} = \sum_{t' \leq t} \left( \bar{\mathbf{c}}^{(t)} \prod_{s=t'+1}^{t} \mathbf{J}^{(s)} \right)_i \alpha\phi'(v_i^{(t')})\hat{r}_j^{(t'-1)}, \tag{1}$$

33rd Conference on Neural Information Processing Systems (NeurIPS 2019), Vancouver, Canada.

| Name | Tensor(s) | Update equations | Notes |
|---|---|---|---|
| UORO [9] | $A_k^{(t)} B_{ij}^{(t)}$ | $A_k^{(t)} = \rho_0 \sum_{k'} J_{kk'}^{(t)} A_{k'}^{(t-1)} + \rho_1 \nu_k$ 
 $B_{ij}^{(t)} = \rho_0^{-1} B_{ij}^{(t-1)} + \rho_1^{-1} \nu_i \phi'(v_i^{(t)}) \hat{r}_j^{(t-1)}$ | $\boldsymbol{\nu}$ i.i.d. $\sim \{\pm 1\}$, $\rho_0, \rho_1 > 1$ constants, can be chosen to minimize variance |
| KF-RTRL [10] | $A_j^{(t)} B_{ki}^{(t)}$ | $A_j^{(t)} = \nu_0 \rho_0 A_j^{(t-1)} + \nu_1 \rho_1 \hat{r}_j^{(t-1)}$ 
 $B_{ki}^{(t)} = \nu_0 \rho_0^{-1} \sum_{k'} J_{kk'}^{(t)} B_{k'i}^{(t-1)} + \nu_1 \rho_1^{-1} \alpha \delta_{ki} \phi'(v_i^{(t)})$ | |
| R-KF [14] | $A_i^{(t)} B_{kj}^{(t)}$ | $A_i^{(t)} = \rho_0 A_i^{(t-1)} + \rho_1 \nu_i$ 
 $B_{kj}^{(t)} = \rho_0^{-1} \sum_{k'} J_{kk'}^{(t)} B_{k'j}^{(t-1)} + \rho_1^{-1} \nu_i \phi'(v_i^{(t)}) \hat{r}_j^{(t-1)}$ | |
| KeRNL [11] | $A_{ki}^{(t)} B_{ij}^{(t)}$ | $A_{ki}^{(t)} = A_{ki}^{(t-1)} - \eta_A \left( \sum_{i'} A_{ki'}^{(t-1)} \zeta_{i'}^{(t)} - \Delta r_k^{(t)} \right) \zeta_i^{(t)}$ 
 $B_{ij}^{(t)} = (1 - \alpha_i) B_{ij}^{(t-1)} + \alpha_i \phi'(v_i^{(t)}) \hat{r}_j^{(t-1)}$ | $\Delta r_k^{(t)}$ noise effects, $\zeta_i^{(t)}$ filtered noise 
 $\eta_A$ learning rate, $\alpha_i$ learned timescales |
| RFLO [12] | $\delta_{ki}^{(t)} B_{ij}^{(t)}$ | $B_{ij}^{(t)} = (1 - \alpha) B_{ij}^{(t-1)} + \alpha \phi'(v_i^{(t)}) \hat{r}_j^{(t-1)}$ | $\alpha$ timescale of forward pass |
| DNI [13] | $A_{li}^{(t)}$ | $A_{li}^{(t)} = A_{li}^{(t-1)} - \eta_A \left[ \sum_{l'} \tilde{r}_{l'}^{(t)} A_{l'i}^{(t-1)} - c_i^{*(t)} \right]$ 
 where $c_i^{*(t)} = \bar{c}_i^{(t)} + \sum_m \sum_{l'} \tilde{r}_{l'}^{(t+1)} A_{l'm}^{(t-1)} J_{mi}^{(t+1)}$ | $l$ indexes entries of $\tilde{\mathbf{r}}^{(t)} = [\mathbf{r}^{(t)}; \mathbf{y}^{*(t)}; 1]$. |

Table 1: A summary of several new online algorithms' tensor structure and update equations.

where $\bar{c}^{(t)} \equiv \partial L^{(t)} / \partial \mathbf{r}^{(t)} \in \mathbb{R}^n$ is the immediate credit assignment vector and $\mathbf{J}^{(s)} \equiv \partial \mathbf{r}^{(s)} / \partial \mathbf{r}^{(s-1)}$ is the network Jacobian, with elements $J_{ij}^{(s)} = (1 - \alpha)\delta_{ij} + \alpha\phi'(h_i^{(s)})W_{ij}^{\text{rec}}$. While Eq. (1) explicitly references activity at all time points, RTRL instead recursively updates the "influence tensor" $M_{kij}^{(t)} = \partial r_k^{(t)} / \partial W_{ij}$ by $M_{kij}^{(t)} = \sum_{k'} J_{kk'}^{(t)} M_{k'ij}^{(t-1)} + \alpha\delta_{ki}\phi'(v_i^{(t)})\hat{r}_j^{(t-1)}$, preserving the first-order long-term dependencies in the network as it runs forward. The actual gradient is then calculated as

$$\frac{\partial L^{(t)}}{\partial W_{ij}} = \sum_k \frac{\partial L^{(t)}}{\partial r_k^{(t)}} \frac{\partial r_k^{(t)}}{\partial W_{ij}} = \sum_k \bar{c}_k^{(t)} M_{kij}^{(t)}. \qquad (2)$$

Unlike BPTT, every computation in RTRL involves only current time $t$ or $t-1$. In general, an online algorithm has some **tensor structure** for summarizing the inter-temporal dependencies in the network, to avoid having to explicitly unroll the network. These tensor(s) must **update** at each time step as new data come in. RTRL uses an order-3 tensor, resulting in an $\mathcal{O}(n^3)$ memory requirement that is neither efficient nor biologically plausible. However, all of the new online algorithms we discuss are only $\mathcal{O}(n^2)$ in memory. In Table 1 we show the tensor structure and update equations for each of these algorithms in order to discuss the mathematical operations needed for each and whether a neural circuit could implement them. How these updates lead to sensible learning is outside our scope, and we refer the reader to either the original papers [9, 10, 12, 11, 13] or the review [14].

In a purely artificial setting, these tensor updates from Table 1 are straightforward to implement, but biologically, one has to consider how these tensors are physically represented and the mechanism for performing the updates. We present a list of mathematical operations and comment on how a biological neural network might or might not be able to perform it in parallel with the forward pass:

 i A vector can be encoded as a firing rate, voltage, or any other intracellular variable.
 ii A matrix must be encoded as the strengths of a set of synapses; if individual entries change, they must do so time-continuously and via a (local) synaptic plasticity rule.
 iii Matrix-vector multiplication can be implemented by neural transmission, but input vectors must represent firing rates, as opposed to voltages or other intracellular variables.
 iv Matrix-matrix multiplication is at face value not possible, as it requires $\mathcal{O}(n^3)$ multiplications, and there is no biological structure to support this.
 v Independent additive noise is feasible; biological neural networks are naturally noisy in ways that can be leveraged for computation.
 vi At face value, it is not possible to maintain a "noisy" copy of the network to estimate perturbation effects, e.g. KeRNL (Table 1) or [15]. However, there may be workarounds.

**How do different algorithms do?** RFLO is sufficiently simple to pass all of these tests, but it arguably doesn't actually solve temporal credit assignment and merely regresses natural memory traces to task labels (see Section 5.5 of [14]), which limits its performance ceiling. Every other algorithm fails at least one of our criteria, at least at first glance. KF-RTRL and R-KF are out because of the matrix-matrix products in their updates. Although the eligibility-trace-like update in KeRNL for $B_{ij}^{(t)}$ is straightforward, learning the $A_{ki}^{(t)}$ matrix requires a perturbed network–on the surface unlikely biologically (vi). While UORO uses only matrix-vector products, the time-continuity requirement (ii) is awkward, because if we choose the constants $\rho_0, \rho_1$ to make one update equation smooth in time (e.g. $\rho_0 = 1 - \epsilon, \rho_1 = \epsilon$, for $0 < \epsilon \ll 1$), the other update becomes unstable due to the appearance of $\rho_0^{-1}, \rho_1^{-1}$. DNI avoids matrix-matrix

Figure 1: **a)** Cross-entropy loss for networks trained on Add task with $\alpha = 0.5$, $t_1 = 3$, and $t_2 = 5$ for various algorithms. Lines are means over 20 random seeds (weight init. and training examples), and shaded regions represent $\pm 1$ S.E.M. Raw loss curves are first down-sampled by a factor of $10^{-4}$ (rect. kernel) and then smoothed with a 10-time-step running average. **b)** Same for mean-squared error on Mimic task.

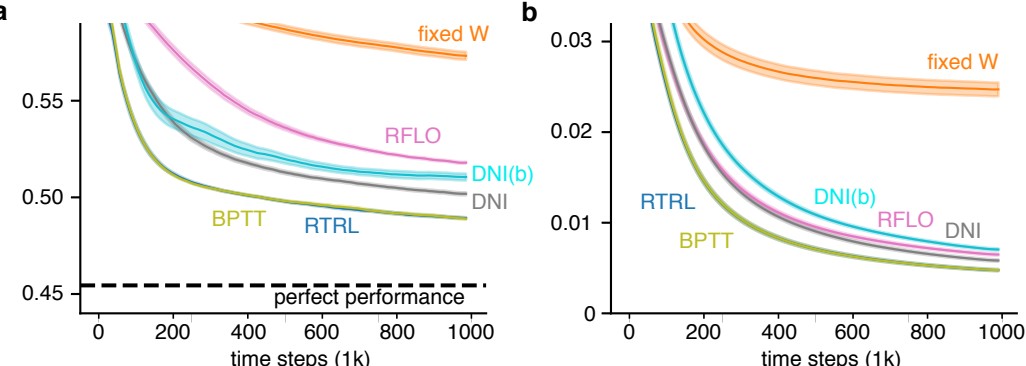

products and is naturally time-continuous, but it involves a matrix-matrix-vector product, requiring physical storage of the intermediate vector. To top it off, every algorithm (except RFLO) requires the Jacobian of the network, but superficially this is not possible (ii), because a set of synapses cannot instantaneously readjust to their ideal values $J_{ij}^{(t)} = (1 - \alpha)\delta_{ij} + \alpha\phi'(v_i^{(t)})\hat{r}_j^{(t-1)}$.

**Can we fix any of these issues?** While each algorithm poses its own challenges, the Jacobian is a recurring problem for anything that meaningfully solves credit assignment. Therefore we propose a general solution, to instead use an *approximate* Jacobian, whose entries we refer to as $\mathcal{J}_{ij}$, which updates at each step according to a perceptron-like learning rule:

$$\Delta\mathcal{J}_{ij} \propto -(r_i^{(t)} - \sum_{j'} \mathcal{J}_{ij'} r_{j'}^{(t-1)}) r_j^{(t-1)}. \tag{3}$$

Biologically, this would correspond to having an additional set of synapses (possibly spatially segregated from $\mathbf{W}$) with their own plasticity rules [16]. Computationally, this approximation brings no traditional speed benefits, but it offers a plausible mechanism by which a neural circuit can access its own Jacobian for learning purposes.

As for other challenges, the matrix-matrix-vector product appearing in DNI can be implemented by the circuit itself in a phase of computation separate from the forward pass. For the intermediate result to pass through the second matrix, it must be represented as a firing rate (iii), which already requires altering the original equations to $\sum_m \phi\left(\sum_{l'} \tilde{r}_{l'}^{(t+1)} A_{l'm}\right) J_{mi}$, since on its own $u_m^{(t+1)} \equiv \sum_{l'} \tilde{r}_{l'}^{(t+1)} A_{l'm}$ is a voltage. This would naively interfere with the forward pass, since $\mathbf{v}^{(t)} = \mathbf{W}\hat{\mathbf{r}}^{(t-1)}$ already uses the network firing rates and somatic voltages. However, we could imagine the $\mathbf{A}$ synapses feeding into an electrically isolated neural compartment (say the apical dendrites) to define a separate voltage $u_m^{(t+1)}$, which is allowed to drive neural firing to $\phi(u_m^{(t+1)})$ in specific "update" phases. We already know that branch-specific gating (by interneurons) can filter which information makes it to the soma to drive spiking [17].

**Do these fixes work empirically?** Given our criteria and novel workarounds, RFLO and DNI(b), our altered version of DNI (with the approximate Jacobian), remain as viable candidates for neural learning. To ensure our additional approximations do not ruin performance, we empirically evaluate DNI(b), along with the original DNI and RFLO. As upper and lower bounds on performance, respectively, we also include exact credit assignment methods (BPTT and RTRL) and a "fixed-$\mathbf{W}$" algorithm that only trains the output weights. We use two synthetic tasks, each of which requires solving temporal credit assignment and has clear markers for success. One task ("Add") requires mapping a stream of i.i.d. Bernoulli inputs $x^{(t)}$ to an output $y^{*(t)} = 0.5 + 0.5x^{(t-t_1)} - 0.25x^{(t-t_2)}$ [18], with time rescaled to match $\alpha$. The label depends on the inputs via lags $t_1, t_2$ that can be adjusted to modulate task difficulty. The other task ("Mimic") requires reproducing the response of a separate RNN with the same architecture and fixed weights to a shared Bernoulli input stream. We find that training loss for RFLO and DNI is worse than the optimal solutions (BPTT and RTRL), but both beat the fixed-$\mathbf{W}$ performance lower bound. DNI(b) performs worse than original DNI, unsurprising because it involves further approximations, but still much better than the fixed-$\mathbf{W}$ baseline. This demonstrates that solving temporal credit assignment is possible within biological constraints.

### Discussion

It is still unclear how neural circuits achieve sophisticated learning, in particular solving temporal credit assignment. Here we approached the problem by looking for biologically sensible approximations to RTRL and BPTT. Although we have empirical results to prove that our solutions can solve temporal credit assignment for simple tasks, the substance of our contribution is conceptual, in articulating what computations are abstractly feasible and which are not. In particular, we have shown that accessing the Jacobian for learning is possible by using a set of synapses trained to linearly approximate the network's own dynamics.

Along the way, we have identified some key lessons. The main one is that neural circuits need additional infrastructure specifically to support learning. This could be extra neurons, extra compartments within neurons, separate coordinated phases of computation, input

gating by inhibition, etc. While we all know that biology is a lot more complicated than traditional models of circuit learning would suggest, it has proved difficult to identify the functional role of these details in a bottom-up way. On the other hand, drawing a link between ML algorithms and biology can hint at precise computational roles for not well understood circuit features.

Another lesson is that implementing even fairly simple learning equations in parallel to the forward pass is nontrivial, since it already uses up so much neural hardware. Even a simple matrix-vector product requires an entirely separate phase of network dynamics in order to not interfere with the forward pass of computation. While it may be tempting to outsource some of these update equations to separate neurons, the results would not be locally available to drive synaptic plasticity.

Of course, we acknowledge that any particular solution, whether RFLO or DNI, is a highly contrived, specific, and likely incorrect guess at how neural circuits learn, but we believe the exercise has big-picture implications for how to think about biological learning. Beyond the particular topic of online learning in recurrent networks, our work provides a general blueprint for abstractly evaluating computational models as mechanistic explanations for biological neural networks. Knowing what computational building blocks are at our disposal and what biological details are needed to implement them is an important foundation for studying ML algorithms in a biological context.

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

## Appendix A: Algorithm Details

### The "Stochastic Family"

The first three algorithms in Table 1—Unbiased Online Recurrent Optimization (UORO, [9]), Kronecker-Factored Real-Time Recurrent Learning (KF-RTRL, [10]), and Reverse-KF-RTRL (R-KF, [14])—all make use of explicit randomness to efficiently approximate the influence tensor $M_{kij}^{(t)}$ via an unbiased estimate built from lower-rank tensors. For example, UORO approximates $M_{kij}^{(t)}$ as $A_k^{(t)} B_{ij}^{(t)}$, KF-RTRL as $A_j^{(t)} B_{ki}^{(t)}$. Of course, the whole point is to avoid the full multiplication that would approximate $M_{kij}^{(t)}$ and instead update each component tensor individually. Attempting to do so in a way that implements the update $M_{kij}^{(t)} = \sum_{k'} J_{kk'}^{(t)} M_{k'ij}^{(t-1)} + \alpha \delta_{ki} \phi'(v_i^{(t)}) \hat{r}_j^{(t-1)}$ will necessarily produce unwanted terms. The key "trick" common to all 3 algorithms (originally proposed in [19]) is that, by using a vector of random i.i.d. $\nu_k \in \{\pm 1\}$ in the updates of $A$ and $B$, we can recover the RTRL update while unwanted cross-terms vanish in expectation. We show explicitly how the update for UORO produces an unbiased estimate of $M_{kij}^{(t)}$ (other cases are similar). Writing the "immediate influence" $\overline{M}_{kij}^{(t)} \equiv \alpha \delta_{ki} \phi'(v_i^{(t)}) \hat{r}_j^{(t-1)}$ for convenience, we have

$$
\begin{aligned}
A_k^{(t)} B_{ij}^{(t)} &= \left( \rho_0 \sum_{k'} J_{kk'}^{(t)} A_{k'}^{(t-1)} + \rho_1 \nu_k \right) \left( \rho_0^{-1} B_{ij}^{(t-1)} + \rho_1^{-1} \sum_{k'} \nu_{k'} \overline{M}_{k'ij}^{(t)} \right) \\
&= \sum_{k'} J_{kk'}^{(t)} A_{k'}^{(t-1)} B_{ij}^{(t-1)} + \sum_{k'} \nu_k \nu_{k'} \overline{M}_{k'ij}^{(t)} + \sum_{k'} \nu_{k'} \left[ \rho_1 \rho_0^{-1} \delta_{kk'} B_{ij}^{(t-1)} + \rho_0 \rho_1^{-1} \overline{M}_{k'ij}^{(t)} \sum_{k''} J_{k'k''}^{(t)} A_{k''}^{(t-1)} \right] \\
\implies \mathbb{E}\left[ A_k^{(t)} B_{ij}^{(t)} \right] &= \sum_{k'} J_{kk'}^{(t)} \mathbb{E}\left[ A_{k'}^{(t-1)} B_{ij}^{(t-1)} \right] + \sum_{k'} \mathbb{E}[\nu_k \nu_{k'}] \overline{M}_{k'ij}^{(t)} + \sum_{k'} \mathbb{E}[\nu_{k'}] \text{ (cross terms)} \\
&= \sum_{k'} J_{kk'}^{(t)} M_{k'ij}^{(t-1)} + \sum_{k'} \delta_{kk'} \overline{M}_{k'ij}^{(t)} + \sum_{k'} 0 \times \text{(cross terms)} \\
&= \sum_{k'} J_{kk'}^{(t)} M_{k'ij}^{(t-1)} + \overline{M}_{kij}^{(t)} = M_{kij}^{(t)}
\end{aligned}
$$

by inductively assuming that the estimate is unbiased, i.e. $\mathbb{E}\left[ A_{k'}^{(t-1)} B_{ij}^{(t-1)} \right] = M_{k'ij}^{(t-1)}$.

### The "E-trace" Family

Both KeRNL and RFLO leverage eligibility-trace-like terms $B_{ij}^{(t)}$ that temporally filter the immediate influences $\alpha \phi'(v_i^{(t)}) \hat{r}_j^{(t-1)}$ at each synapse with learned forgetting factors $1 - \alpha_i$. KeRNL additionally trains (via perturbation methods) a matrix $A_{ki}$ to represent the long-term sensitivity:

$$
\frac{\partial a_k^{(t)}}{\partial a_i^{(t')}} \approx A_{ki} (1 - \alpha_i)^{(t - t')}. \tag{4}
$$

By fixing $t = t'$ and using the chain rule, two special cases of this equation emerge: $A_{ki} \approx \delta_{ki}$ and $A_{ki} \approx (1 - \alpha_i)^{-1} \sum_{k'} J_{kk'}^{(t)} A_{k'i}$. Then the update for $B_{ij}^{(t)}$ effectively drives the RTRL update:

$$
\begin{aligned}
A_{ki} B_{ij}^{(t)} &= A_{ki} \left[ (1 - \alpha_i) B_{ij}^{(t-1)} + \alpha \phi'(h_i^{(t)}) \hat{a}_j^{(t-1)} \right] \\
&= A_{ki} (1 - \alpha_i) B_{ij}^{(t-1)} + \alpha A_{ki} \phi'(h_i^{(t)}) \hat{a}_j^{(t-1)} \\
&\approx \sum_{k'} J_{kk'}^{(t)} A_{k'i} B_{ij}^{(t-1)} + \alpha \delta_{ki} \phi'(h_i^{(t)}) \hat{a}_j^{(t-1)} \\
&\approx \sum_{kk'} J_{kk'}^{(t)} M_{k'ij}^{(t-1)} + \overline{M}_{kij}^{(t)} = M_{kij}^{(t)},
\end{aligned}
$$

where we used both approximations for $A_{ki}$ and inductively assumed $A_{ki} B_{ij}^{(t-1)} \approx M_{kij}^{(t-1)}$.

### DNI

DNI [13] is fundamentally unlike the others, in that it does not approximate the "past-facing" gradient $\partial L^{(t)} / \partial \mathbf{W}$ calculated by RTRL, but rather a "future-facing" gradient $\partial \mathcal{L} / \partial \mathbf{W}^{(t)}$, where $\mathbf{W}^{(t)}$ indicates the specific application of the recurrent parameters at time $t$, and $L^{(t)}$ the loss at time $t$ (see [14]). The credit assignment vector $\mathbf{c}^{(t)} \equiv \partial \mathcal{L} / \partial \mathbf{r}^{(t)}$, rather than calculated exactly using backpropagation, is *estimated* by a linear function (called the "synthetic gradient") of $\tilde{\mathbf{r}}^{(t)} = \text{concat}(\mathbf{r}^{(t)}; \mathbf{y}^{*(t)}; 1)$, i.e. $\mathbf{c}^{(t)} \approx \tilde{\mathbf{r}}^{(t)} \mathbf{A}$. The matrix $\mathbf{A}$ is ideally learned by gradient descent on the loss function $L_{SG}^{(t)} = \frac{1}{2} ||\tilde{\mathbf{r}}^{(t)} \mathbf{A} - \mathbf{c}^{(t)}||^2$—that is, trained to match the true credit assignment vector—but since the whole point is to avoid calculating $\mathbf{c}^{(t)}$ exactly, the label is replaced by a bootstrapped estimate $\overline{\mathbf{c}}^{(t)} + \tilde{\mathbf{r}}^{(t+1)} \mathbf{A} \mathbf{J}^{(t+1)} \approx \overline{\mathbf{c}}^{(t)} + \mathbf{c}^{(t+1)} \mathbf{J}^{(t+1)} = \mathbf{c}^{(t)}$, where the last equality is by the chain rule (see either [13] or [14] for further detail). This bootstrapped estimate uses the same $\mathbf{A}$ matrix on $\mathbf{c}^{(t+1)}$.

