# OpenReview forum: "Evaluating biological plausibility of learning algorithms the lazy way"
_NeurIPS.cc/2019/Workshop/Neuro_AI — Real Neurons & Hidden Units @ NeurIPS 2019 Poster_

### Official Review · AnonReviewer2 · 2019-09-17
**Interesting analysis of biologically plausible recurrent network learning**

**Clarity:** 5

**Comment:**

Additional diagrams and perhaps a short summary of each learning algorithm would help for the final submission. The authors could also discuss/speculate how these ideas might map on to specific circuits in cortex/hippocampus/etc.

Overall, great work!

**Category:**

Common question to both AI & Neuro

**Clarity Comment:**

Overall, the submission is very clear. For the final submission, the authors could improve the clarity even further by elaborating on their findings/set-up and including diagrams of learning algorithms/techniques.


**Evaluation:**

5: Excellent

**Importance:**

4: Very important

**Importance Comment:**

The submissions has a number of important contributions: 1) suggesting a list of criteria for evaluating biologically plausible learning algorithms, 2) comparing the biological plausibility of recently proposed real time recurrent learning algorithms, and 3) proposing and evaluating a method for approximating the network Jacobian online.

**Intersection:**

4: High

**Intersection Comment:**

This submission includes aspects of both neuroscience and machine learning. The findings may be more relevant to a neuroscience audience, but members from both fields will find the work interesting and insightful.

**Rigor Comment:**

The technical rigor is superb. Mathematical terms are all properly defined, algorithms are defined in these terms, and the new approximation method is empirically evaluated on some simple tasks.

**Technical Rigor:**

4: Very convincing

---

> ### Author Response · Authors · 2019-10-29
> **Reviewer 2 response**
>
> Thanks for your comments, especially about clarity. We used the extra page to summarize each algorithm as succinctly as possible. As to the hippocampal/cortical circuit analogies, the model is admittedly not easy to relate to circuit-specific biological detail. At the abstract level, it does make some predictions about the roles of different dendritic compartments during learning (e.g. distal modulation of basal plasticity, Dudman et al. 2007). We know hippocampal and cortical circuits have different organizing principles for which dendrites are driven by external inputs, which might help make predictions more specific.

---

### Official Review · AnonReviewer3 · 2019-09-20
**Great submission, exactly the sort of approach the field needs**

**Clarity:** 4

**Comment:**

Fantastic submission, perfect for the workshop. I look forward to seeing it presented!

**Category:**

AI->Neuro

**Clarity Comment:**

The paper is very well written, but, possibly due to space constraints, it was a bit hard to follow all the various algorithms discussed. On that note: it would be better to cite the original papers in table 1, so readers can look them up and compare without having to check back through the text.

**Evaluation:**

5: Excellent

**Importance:**

5: Astounding importance

**Importance Comment:**

The current field of biologically plausible learning rules is littered with many proposals and few unifying frameworks. This paper addresses that nicely by breaking down the specific elements required for temporal credit assignment and assessing the biological plausibility of each one. This is an immensely important change in tact that the field needs more of!

**Intersection:**

5: Outstanding

**Intersection Comment:**

It is right at the intersection of ML and comp neuro.

**Rigor Comment:**

Overall, the paper is technically excellent. There are some lingering questions I have about potential means of implementing Jacobians biologically, and a few other minor things, but overall the framework is very clear and the arguments well founded. The demonstration of the learning capabilities of the modified DNI rule is great as well.

**Technical Rigor:**

4: Very convincing

---

> ### Author Response · Authors · 2019-10-29
> **Reviewer 3 response**
>
> Thank you. Excellent point about including citations in the table—we have done so in the camera-ready version. We agree that the description of the algorithms was very dense, and we have used the additional page to explain the principles behind each algorithm as briefly as possible.

---

### Official Review · AnonReviewer1 · 2019-09-23
**A necessary beginning to systematic evalulation of ML algorithms in terms of bioplausibility**

**Clarity:** 4

**Comment:**

The authors provide a very nice, principled survey of several AI algorithms in terms of biological plausibility, focusing specifically on biologically plausible ways to implement operations involving the network Jacobian. While the authors didn't strongly suggest any novel algorithms as a result (besides DNI(b) ), this is nonetheless a useful first step toward establishing a common framework for developing new approaches in both neuroscience and AI.

One thing I think would have been useful to mention, even if rigorous analysis was beyond the scope of the manuscript, would be unsupervised and reinforcement learning algorithms, in which errors are not necessarily defined by moment-to-moment differences between generated and target time-series, but rather in terms of sporadic rewards and punishment, and which may have a deeper intrinsic link to biological learning rules.

**Category:**

AI->Neuro

**Clarity Comment:**

While it was clear in Table 1 which algorithms required e.g. the network Jacobian or matrix products, its presentation could have probably been simplified quite a bit. Given the large number of different mathematical ideas they needed to convey, however, the paper was generally quite straightforward to read.

**Evaluation:**

4: Very good

**Importance:**

4: Very important

**Importance Comment:**

To understand the potential for various learning rules in artificial neural networks in terms of biological plausibility the authors enumerate specific criteria to systematically evaluate bioplausibility in several state-of-the-art learning algorithms. While mostly a principled survey of existing algorithms rather than new research results, this is nonetheless an important step forward in clarifying the relationships between artificial intelligent systems and the brain.

**Intersection:**

5: Outstanding

**Intersection Comment:**

This paper directly evaluates several AI learning algorithms in terms of their biological plausibility.

**Rigor Comment:**

Although space-limited, the authors did a nice job on emphasizing the key computational features of the learning context and specific algorithms they explored, without glossing over mathematical details. The specific enumeration of bioplausibility criteria, while written in words, also nicely provided a principled mathematical basis for their analyses.

**Technical Rigor:**

4: Very convincing

---

> ### Author Response · Authors · 2019-10-29
> **Reviewer 1 response**
>
> Thank you for the thoughtful comments. We agree that learning via sparse rewards and punishments is a core issue for linking biological and artificial learning, and the deep reinforcement learning techniques (e.g. deep Q learning) don’t seem biologically plausible on the surface. Our emphasis here is that, even if such the sparse rewards and punishments could be translated into useful error signals for the agent’s output, propagating these signals over time is challenging biologically at the implementation level.

---

### Decision · Program_Chairs · 2019-10-02

Accept (Poster)